# Efficacy of Liposomal Melatonin in sleep EEG in Childhood: A Double Blind Case Control Study

**DOI:** 10.3390/ijerph20010552

**Published:** 2022-12-29

**Authors:** Alice Bonuccelli, Andrea Santangelo, Francesca Castelli, Giulia Magherini, Elena Volpi, Elisa Costa, Elena Alesci, Gabriele Massimetti, Francesca Felicia Operto, Diego Giampiero Peroni, Alessandro Orsini

**Affiliations:** 1Pediatric Neurology, University Hospital of Pisa, Azienda Ospedaliero Universitaria Pisana, 56126 Pisa, Italy; 2Pediatric Clinic, University Hospital of Pisa, Azienda Ospedaliero Universitaria Pisana, 56126 Pisa, Italy; 3Department of Clinical and Experimental Medicine, University of Pisa, Azienda Ospedaliero Universitaria Pisana, 56126 Pisa, Italy; 4Child and Adolescent Neuropsychiatry Unit, Department of Medicine, Surgery and Dentistry, University of Salerno, 84081 Baronissi, Italy

**Keywords:** EEG, melatonin, neurophysiology, sleep EEG

## Abstract

Electroencephalography (EEG) is pivotal in the clinical assessment of epilepsy, and sleep is known to improve the diagnostic yield of its recording. Sleep-EEG recording is generally reached by either partial deprivation or by administration of sleep-inducing agents, although it is still not achieved in a considerable percentage of patients. We conducted a double-blind placebo-controlled study, involving a hundred patients between 1 and 6 years old, randomized into two groups: Group 1 received liposomal melatonin (melatosome) whereas Group 2 received a placebo. Sleep latency (SL), defined as the time span between the onset of a well-established posterior dominant rhythm, considered as a frequency of 3 to 4 Hz, increasing to 4–5 Hz by the age of 6 months, to 5–7 Hz by 12 months, and finally to 8 Hz by 3 years, and the first EEG sleep figures detected, were measured for each patient. A significant difference in SL was observed (10.8 ± 5 vs. 18.1 ± 13.4 min, *p*-value = 0.002). Within each group, no differences in sleep latency were detected between genders. Furthermore, no difference in EEG abnormality detection was observed between the two groups. Our study confirmed the efficacy and safety of melatonin administration in sleep induction. Nonetheless, liposomal melatonin presents a greater bioavailability, ensuring a faster effect and allowing lower dosages. Such results, never before reported in the literature, suggest that the routine employment of melatonin might improve clinical practice in neurophysiology, reducing unsuccessful recordings.

## 1. Introduction

Electroencephalography (EEG) is a pivotal diagnostic tool in the clinical assessment of a first seizure episode in children for a differential diagnosis between seizure and seizure-like attacks, and as a necessary diagnostic instrument in monitoring epilepsy evolution [1,2].

However, only 50% of epileptiform abnormalities are detected during a standard, awake recording [3]. Sleep is known to improve the diagnostic yield of the EEG recording, particularly in children. A sleep EEG greatly increases the amount of diagnostic evidence compared to an awake EEG, minimizing the artifacts linked to poor cooperation [4,5], leading to important data on the brain’s electrical activity development, and lowering the epilepsy threshold [6].

Sleep deprivation before EEG recording seems to increase both the chance of sleep recording during the procedure and the detection of interictal epileptiform discharges [7]: epileptiform abnormalities are, in fact, generally triggered by sleep, which therefore is recommended in order to achieve a more precise definition of epileptic illnesses [2,7].

Sleep in children is generally reached by either partial deprivation or by the administration of sleep-inducing drugs [8,9]. Interestingly, sleep deprivation has only a modest influence on the macrostructure of sleep, with a small increase in the duration of stage 2 NREM [4]. On the other hand, sleep deprivation is perceived as troublesome and stressful both by caregivers and children [10,11,12,13]. Although caregivers are regularly trained to sleep deprive their children prior to their scheduled exam, spontaneous sleep is still not achieved in a considerable percentage of patients (approximately 50%), in particular, in those with behavioral disorders and neurodevelopmental delay, which represent a significant percentage of the patients who require EEG [14].

Pharmacological agents, including barbiturates, chlorpromazine, triclofos and chloral hydrate, have all been employed to induce sleep, although most of them can impact the sleep macrostructure and/or affect the analysis of the recorded EEG data [15]. Furthermore, there is also the risk of slowly drifting into a deeper level of unconsciousness, eventually leading to upper airway obstruction and the loss of cough reflex; therefore, close observation and oxygen saturation monitoring are required [16].

Moreover, delayed awakening or persistent sleepiness after the exam could lead to prolonged observation or even hospitalization for safety reasons [17].

Melatonin, an indoleamine which is mostly produced in the pineal gland, has been reported as a promising alternative to achieve sedation in children [18,19]. It has been recognized as a “ubiquitously distributed and functionally diverse molecule”. In fact, melatonin is involved in several physiological functions, including the modulation of circadian rhythm and regulation of season changes, and it presents antioxidant, anti-inflammatory, oncostatic, antimigraine and anticonvulsant properties [19,20]. Of note, the anticonvulsant effects might be linked to an inhibition of GABAergic transmission which only occurs at very high dosages, and has therefore no clinical utility.

Exogenous melatonin is also an advantageous oral natural-sleep inducer, currently employed in the treatment of sleep disorders and jet lag [21,22]. Its efficiency and safety in sleep induction for noninvasive procedures, both in adults and children (including those with developmental or behavioral impairment), have been successfully demonstrated [23]. Furthermore, parental compliance is usually enhanced in the case of administration of molecules that are originally synthesized by the organism [8,24].

In most studies, melatonin is administered at dosages ranging between 5 and 20 mg/day (also in repeated doses), in the absence of significant adverse effects, confirming its excellent pharmacological profile.

We conducted a double-blind placebo-controlled study, in order to evaluate the efficacy of liposomal melatonin as a sleep inducer in pediatric patients performing a polygraphic electroencephalograph (pEEG) during sleep. The aim of the present study was to assess the efficacy in sleep induction of liposomal melatonin against a placebo; the future application is the introduction of liposomal melatonin in clinical practice in pediatric EEG. We also evaluated the safety of liposomal melatonin, closely monitoring the occurrence of eventual adverse events.

## 2. Materials and Methods

Participants were randomized into a two-group, parallel, placebo-controlled double blind trial. Neither the experimenter nor the patients’ caregivers knew to which group they were randomized.

Group 1 (G1) received liposomal melatonin (melatosome), which was administered in two different doses: 3 mg (0.75 mL) for children aged 1 to 3 years old, and 5 mg (1.25 mL) for children between 4 and 6 years old. Group 2 (G2), on the other hand, received placebo in vials of 0.75 mL and 1.25 mL, administered, respectively, to the same age groups as G1. The two product were visually identical.

All our patients were sleep deprived, according to the current guidelines for performing EEG in sleep. Namely, all children were awakened at 6.00 am; in addition, patients from two to four years old were kept up for additional hours past their bedtime, whereas older children kept up until midnight. Electrodes were arranged through an EEG cap according to Jasper 10–20 system.

We included patients between 1 and 6 years old assessed at the Pediatric Neurologic Clinic of Pisa and equipped with a referral for an EEG in wakefulness and sleep. Exclusion criteria included the denial of informed consent, a previously diagnosed sleep disorder, inability to follow the instructions given by the operator due to communication problems (e.g., lack of knowledge of Italian language), or being unable to take the product.

During the analysis, we discarded patients whose EEG could not be performed, whose parents withdrew informed consent, or who were not compliant enough to consume the product.

After the signature for informed consent was obtained from the caregivers of all eligible children, a randomly assigned vial was administered to the patient, and the EEG was recorded at the sleep lab of our Department.

Subsequently, two independent pediatric neurologist experts in epilepsy and EEG evaluated the EEG tracking and determined the sleep latency (SL) for each patient. SL was defined as the time span between the onset of a well-established posterior dominant rhythm and the first EEG sleep figures detected (sleep spindles, K complexes).

The primary outcome of the study was to evaluate the efficacy of melatonin administration as a sleep-inducer in children who undergo a sleep-EEG examination.

For the statistical analysis of quantitative variables, we evaluated mean, median, and standard deviations, which were compared through *t*-tests and ANCOVA analysis was performed in order to remove the effect of age. To study the relationship between these variables, the Pearson correlation coefficient was calculated. Chi-squared test was employed for the analysis of categorical variables which were expressed as percentages. Sleep latencies were also analyzed with Kaplan–Meier estimator, and compared through Mantel–Cox test. For statistical analysis, we used IBM SPSS, ver. 28.

Randomization was performed with software NCSS 2020 (NCSS 2020 Statistical Software (2020). NCSS, LLC. Kaysville Utah, USA, ncss.com/software/ncss). A balanced randomization for G1 and G2 was obtained with Efron’s procedure (*p* = 0.67). Furthermore, the software generated a randomized univocal code for each patient. All subjects and investigators were blind to treatment conditions until after data analysis.

The present study was conducted according to CONSORT 2010 guidelines.

## 3. Results

Between May 2021 and May 2022, 1492 children accessed our clinic in order for a sleep polygraphic EEG to be performed. Among these, we recruited the first 100 patients who met the criteria of inclusion and exclusion.

Fifty patients received the placebo (control group, CG), whereas fifty received liposomal melatonin (melatonin group, MG).

We evaluated 44 boys and 56 girls, the mean age was 3.7 years (±1.74). Forty-two patients underwent EEG to control a previously diagnosed epileptic syndrome (17 in the CG and 25 in the MG), 12 for febrile seizures (7 in the CG and 5 in the MG), 13 for suspected seizures (7 in the CG and 6 in the MG), 5 for sleep disturbances (1 in the CG and 4 in the MG) and 8 for behavioral problems/psychomotor delay (3 in the CG and 5 in the MG). Thirty-seven patients were treated with an ASM (antiseizure medication), 19 in the MG and the 18 in CG [Figure 1].

The overall average SL was found to be 13.9 ± 10.5 min (range 2–25). A significant difference in SL between the two groups was observed (10.8 ± 5 min in the MG vs. 18.1 ± 13.4 min in the CG, *p*-value = 0.002) (Figure 2 and Figure 3). Within the two groups, no differences in sleep latency were detected between the genders (in males: 20.4 ± 17.4 in the CG vs. 11.5 ± 4.6 in the MG; in girls: 14.5 ± 10.2 in the CG vs. 10 ± 5.6 in the MG), whereas SL appeared to be influenced by age. When the effect of age was removed, the effect of the melatonin use increased and was significant (*p*-value = 0.002). The amount of variation accounted for by melatonin increased to 994.57 (from 827.85) units and the unexplained variance (SSr) was reduced to 9248.05 (from 9634.25). In addition to such results, the Kaplan–Meier curves underlined a significant difference in SL estimation between the two groups (*p*-value: 0.002) (Figure 4). No differences in abnormality detection were observed between the two groups (21 in the CG vs. 16 in the MG, *p*-value = 0.294).

For all 100 patients, no adverse events were observed during the examination, and no patients abandoned the study.

## 4. Discussion

Although difficult to perform, an optimal EEG recording plays a crucial role in the diagnosis and clinical management of alleged epileptic patients. However, sleep deprivation, which, to date, appears to be necessary for sleep-EEG recordings, might represent a difficult struggle, becoming a source of stress for caregivers, who often find sleep deprivation impossible or excessively stressful for their children and themselves [11,12,13].

Melatonin is currently employed in acute and chronic circadian rhythm disorders in children and adolescents, either healthy or with neurologic or neurodevelopmental disorders, such as autism spectrum disorder (ASD) [25].

Although it is reported that sleep duration might be shorter with melatonin, a meta-analysis [26] of 13 pediatric randomized controlled trials revealed that sleep duration was actually increased by approximately 30 min by melatonin administration; however, we believe that such an observation does not apply for EEG-recording purposes, since most EEG recordings usually last less than 30 min [8].

Compared to other sedatives, melatonin administration has shown a similar sleep latency and length of sleep, with a shorter drowsiness period [27,28]. In particular, the efficacy of melatonin was comparable to triclofos, oral midazolam and chloral hydrate, but these are more likely to cause adverse effects. Consistent with these results, the duration of the sleep EEGs in our patients obtained with the administration of melatonin was comparable to those obtained without the administration of melatonin.

Furthermore, our study confirms the efficacy of melatonin administration in sleep induction, and in particular, enhancing its role as an efficient tool in sleep-EEG recording. Furthermore, at the dosages used (3 and 5 mg), we did not find any type of interference in the detection of interictal EEG anomalies. In addition, in our study, we used liposomal melatonin as it has been shown to have a greater bioavailability, which allowed us to use a lower dosage with a faster effect. Such results, which have not previously been reported in the literature, suggest that melatonin could be administered at common dosages for EEG purposes in both epileptic and non-epileptic patients; therefore, we believe that the routine administration of melatonin should be systematically adopted for current practice in neurophysiology, although, to date, there are no guidelines on melatonin administration for sleep-EEG purposes either in adults or pediatric patients.

Given the consistent results of our work, we believe that a combined method (oral melatonin and sleep deprivation) should be adopted prior to EEG recording, at least in specific groups of children, such as patients with behavioral disorders, with inadequate sleep deprivation, or whose parents have not been able to guarantee efficient sleep deprivation.

Of note, our group was able to achieve these results with the administration of the minimum dose of melatonin (3 to 5 mg). We could therefore presume that even better results might be obtained with higher doses. However, safety has yet to be established.

A main limitation of the present study could be represented by the low number of patients. However, our results should encourage further research that might lead to a larger employment of melatonin in neurophysiology. Surprisingly, many parents whose children were recruited for our study spontaneously requested the administration of melatonin for subsequent sleep EEGs, claiming that the examination had been more comfortable for their children.

## 5. Conclusions

In conclusion, ours was the first double-blind, placebo-controlled study that assessed the safety and efficacy of liposomal melatonin as a sleep inducer in pediatric patients undergoing a sleep EEG. We believe that the routine employment of such an indoleamine (liposomal melatonin) might improve clinical practice in neurophysiology by helping to reduce procedural stress in parents and children, enhance the chances of recording sleep during EEG and reduce the number of failed, unsuccessful recordings.

## Figures and Tables

**Figure 1 ijerph-20-00552-f001:**
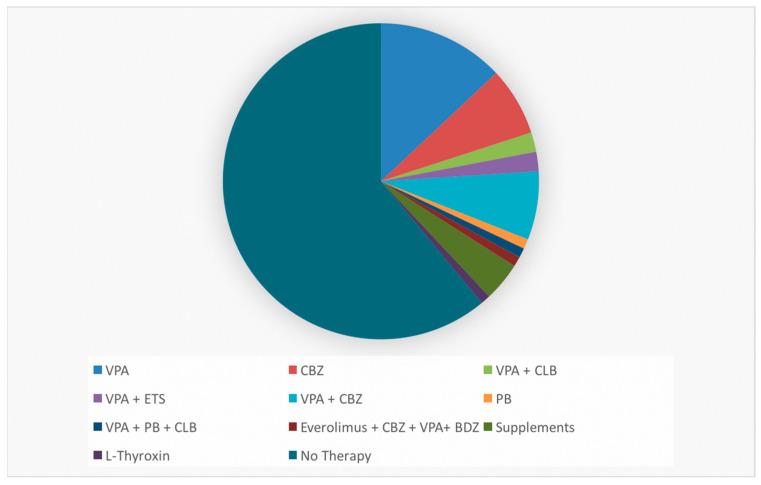
Ongoing therapies at EEG recording in our patients. VPA: valproic acid, ETS: ethosuximide, PB: phenobarbital, CBZ: carbamazepine, CLB: clobazam.

**Figure 2 ijerph-20-00552-f002:**
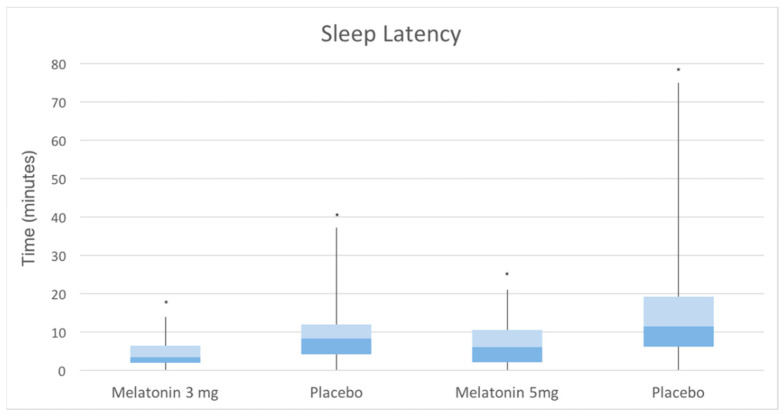
Box plot showing different sleep latencies in MG at different doses and CG. * *p*-value 3 mg vs. placebo: 0.04; 5mg vs. placebo: 0.03.

**Figure 3 ijerph-20-00552-f003:**
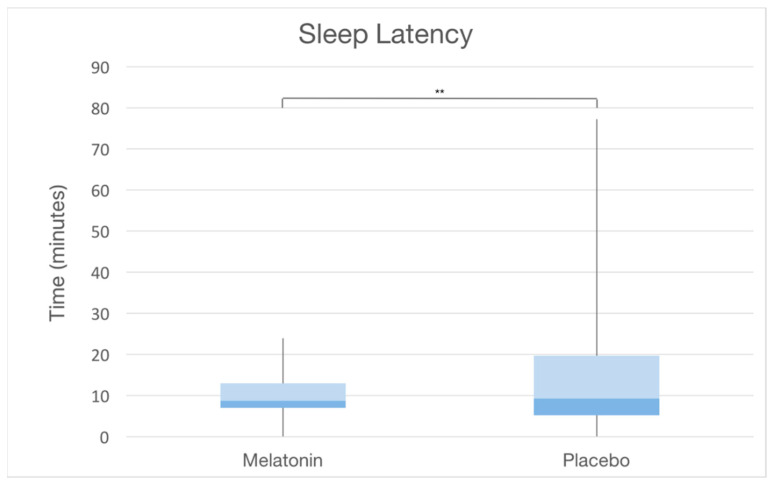
Box plot showing overall differences in sleep latencies between MG and CG. ** *p*-value: 0.002.

**Figure 4 ijerph-20-00552-f004:**
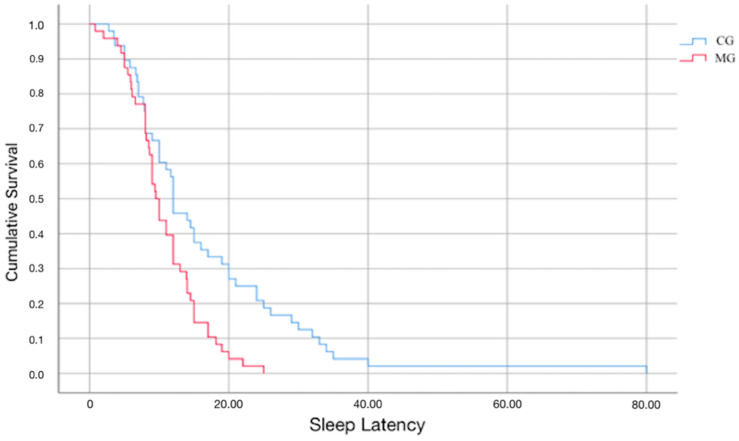
Kaplan–Meier curve for sleep latency in MG and CG.

## Data Availability

The data presented in this study are available on request from the corresponding author. The data are not publicly available due to privacy issues.

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
