# Peer review of "Efficacy of Liposomal Melatonin in sleep EEG in Childhood: A Double Blind Case Control Study"

_ijerph, 2022, doi:10.3390/ijerph20010552_

Round 1

Reviewer 1 Report

The present study is a double blind case control study which aimed to examine the efficacy of Liposomal Melatonin in sleep EEG in childhood. The results seemed that liposomal melatonin could increase the sleep latency and no obvious side effects were observed.The design of the whole study is very simple but very rigorous, making the research conclusions of this study also very reliable.

But there are still some obvious flaws that the author needs to correct.

1. The authors should provide baseline information on the study subjects in addition to therapeutic medication. Moreover, the authors should not simply use the abbreviations for therapeutic medication, although they are familiar to a professional neurologist.

2. It is suggested that the author make appropriate modifications to the figure in the manuscript to make it more beautiful. In addition, additional markers are suggested to indicate the presence of statistical differences. Also, the coordinate axis of the chart represents what must be clearly indicated. Proper figure legend is also needed.

3. Authors should systematically check the writing of the article to avoid some unnecessary errors, such as " For all 100 patients no adverse events were observed during the examination, and no 148 patients abandoned the study " appears twice in the Results section.

4.In the discussion section, the author used too much short paragraphs. It was  advised the author make appropriate mergers to make the expression more fluent.

Author Response

Dear

Ms. Mabel Li

Assistant Editor

Mental Health Section, International Journal of Environmental Research and Public Health

Thank you for the opportunity to resubmit our manuscript “Efficacy of Liposomal Melatonin in sleep EEG in Childhood: a Double Blind Case Control Study”. We sincerely appreciated the Reviewers’ comments, which greatly improved our paper. A point-by-point reply to the Reviewers’ comments is reported below and tracked in the manuscript.

We thank the Reviewer for her/his careful reading of the manuscript and the insightful suggestions

  1. The authors should provide baseline information on the study subjects in addition to therapeutic medication. Moreover, the authors should not simply use the abbreviations for therapeutic medication, although they are familiar to a professional neurologist.

We thank the Reviewer for the advice and specified every abbreviation within the text. The “Results” section has been changes as it follows: “We evaluated 44 boys and 56 girls, mean age was 3.7 years (± 1.74). 42 patients underwent EEG to control a previously diagnosed epileptic syndrome (17 in CG and 25 in MG), 12 for febrile seizures (7 in CG and 5 in MG), 13 for suspected seizures (7 in CG and 6 in MG), 5 for sleep disturbances (1 in CG and 4 in MG) and 8 for behavioural problems/psychomotor delay (3 in CG and 5 in MG). 37 patients were treated with an ASM (antiseizure medication), 19 in MG and 18 in CG.”

  1. It is suggested that the author make appropriate modifications to the figure in the manuscript to make it more beautiful. In addition, additional markers are suggested to indicate the presence of statistical differences. Also, the coordinate axis of the chart represents what must be clearly indicated. Proper figure legend is also needed.

We thank for the suggestion, the figures have been modified in order to ease the reading process, markers have been added to enlighten the statistical difference and legends have been updated.

  1. Authors should systematically check the writing of the article to avoid some unnecessary errors, such as " For all 100 patients no adverse events were observed during the examination, and no 148 patients abandoned the study " appears twice in the Results section.

We apologise for the inconvenient, the manuscript has been revised and redundand concepts have been deleted.

4.In the discussion section, the author used too much short paragraphs. It was  advised the author make appropriate mergers to make the expression more fluent.

We appreciate the suggestion and extensively reviewed the discussion section.

We do hope this revised version will meet your expectations and may now be acceptable for publication in International Journal of Environmental Research and Public Health.

Yours sincerely,

Andrea Santangelo

Paediatric Neurology

Paediatric Department, Santa Chiara University Hospital

Azienda Ospedaliero Universitaria Pisana

Pisa, Italy

Reviewer 2 Report

The authors conducted a double-blind placebo-controlled study, to investigate the efficacy of liposomal melatonin as a sleep inducer in children.

Although the manuscript is well written, and the results are interesting, a lot of extra work needs to be done before it can be published.

General comments:

Only SL is investigated in this study, which is not sufficient. The authors should add more signatures of sleep to evaluate the efficacy of liposomal melatonin on sleep.

For example, the total sleep time, the fragmentation of the sleep, the percentage of different sleep stages across the whole night.

Specific comments:

1, Methods:

(1), More details should be reported about how the subjects were sleep deprived, e.g., how long?

(2), Where the EEG was recorded? At home, at sleep lab? or at the hospital?

(3), How many electrodes are used to identify the SL, or which electrodes are used?

(4), the onset of a well-established posterior dominant rhythm: please be more specific about the rhythm, e.g., frequency range? Amplitude range?

2, Results:

(1),I don't know why did the authors label the first figure as Graphic 1? It should be figure 1, or change it to a table, and name as Table I

(2), Please mark asterisks on figures 1, 2 between groups, which are of significant differences.

(3), Please specify what each y-axis is in figure1 ,2 and 3.

(4), As different age groups received different doses, please give the effect of age on the SL?

(5), What is the difference between the sleep quality of the subjects in different groups? That could also be a factor that will influence the final results.

Author Response

Dear

Mental Health Section, International Journal of Environmental Research and Public Health

Thank you for the opportunity to resubmit our manuscript “Efficacy of Liposomal Melatonin in sleep EEG in Childhood: a Double Blind Case Control Study”. We sincerely appreciated the Reviewers’ comments, which greatly improved our paper. A point-by-point reply to the Reviewers’ comments is reported below and tracked in the manuscript.

Reviewer 2 comments:

We thank the Reviewer for his/her work and further updated our manuscript in order to meet his/her expectations. However, we would like to specify that our results have been observed during an EEG recording in an outpatient setting with a sleep deprivation EEG  of only one hour, as the purpose of our work was to assess the efficacy of melatonin in everyday outpatient activity. Therefore, although intriguing, other data, including the eventual fragmentation of sleep during a whole night could not be assessed. It will certainly be a topic for subsequent publications.

1, Methods:

(1), More details should be reported about how the subjects were sleep deprived, e.g., how long?

We thank the Reviewer for the suggestion and added more informations about sleep deprivation in children. The text has been changed as it follows: “Namely, all children were awakened at 6.00 am, in addition, patients from two to four years-old were kept up for additional hours (2 hours) past their bedtime, whereas older children kept up until midnight”.

(2), Where the EEG was recorded? At home, at sleep lab? or at the hospital?

The EEG was recorded at the hospital in the  pediatric neurophisiology lab of our Unit. We further specified such information in the text (Line 108).

(3How many electrodes are used to identify the SL, or which electrodes are used?

The electrodes employed were applyed though a EEG cap according to Jasper’s 10-20 system.

(4),the onset of a well-established posterior dominant rhythm: please be more specific about the rhythm, e.g., frequency range? Amplitude range?

We considered a posterior dominant rhythm (PDR) as a frequency of 3 to 4 Hz, increasing to 4-5 Hz by the age of 6 months, to 5-7 Hz by 12 months, and finally to 8 Hz by 3 years.

2, Results:

(1),I don't know why did the authors label the first figure as Graphic 1? It should be figure 1, or change it to a table, and name as Table I

We apologise for the inconvenient, figures have been relabeled in order to ease the reading process.

(2), Please mark asterisks on figures 1, 2 between groups, which are of significant differences.

The boxes have been marked to show statistical differences, p-value have also been reported in legends

(3), Please specify what each y-axis is in figure1 ,2 and 3.

We thank for the suggestion, y-axis was labeled in each figure.

(4), As different age groups received different doses, please give the effect of age on the SL?

We thank the Reviewer for the suggestion and further specified the effect of age on the SL of our patients. The section has been modified as it follows: “whereas SL appears to be influenced by the age. When the effect of age is removed, the effect of Melatonin results increased and significant (p-value=0.002). The amount of variation accounted for Melatonin has increased to 994.57 (from 827.85) units and the unexplained variance (SSr) has been reduced to 9248.05 (from 9634.25)”. Such results were obtained through an ANCOVA analysis, which was specified in the Materials and Methods section.

(5), What is the difference between the sleep quality of the subjects in different groups? That could also be a factor that will influence the final results.

Althought the SL has been observed in outpatient setting, a global analysis revealed that patients in MG presented less arousals during the analysis.

We do hope this revised version will meet your expectations and may now be acceptable for publication in International Journal of Environmental Research and Public Health.

Yours sincerely,

Andrea Santangelo

Paediatric Neurology

Paediatric Department, Santa Chiara University Hospital

Azienda Ospedaliero Universitaria Pisana

Pisa, Italy

Round 2

Reviewer 2 Report

The authors have addressed all my comments for this paper and answered my technical questions about this study.